# Examining factors associated with mental health stigma and attitudes toward help-seeking among international college students during the COVID-19 pandemic

**Brandon A. Knettel** [1,2]*, **Pranav Ganapathy** [3], **Conner Rougier-Chapman** [4]

**1** Duke University School of Nursing, Durham, North Carolina, United States of America, **2** Duke Global Health Institute, Durham, North Carolina, United States of America, **3** Cerevel Therapeutics, Cambridge, Massachusetts, United States of America, **4** Wake Forest University, Winston-Salem, North Carolina, United States of America

* brandon.knettel@duke.edu

**Data Availability Statement:** The authors have not received approval from the study's ethical review board, nor the participating academic institutions,

## Abstract

International students in the United States (U.S.) are at increased risk for mental health challenges, but less likely than their U.S.-born peers to seek professional mental health support. We administered an online survey to 132 international students enrolled at 14 U.S. colleges and universities to explore whether demographics, time in the U.S., religiosity, prior contact with people experiencing mental illness, individualism, and collectivism were associated with stigmatizing attitudes and mental health help-seeking. Only increased contact with mental illness was significantly associated with lower mental health stigma in this sample. Identifying as a woman, having more prior contact with mental illness, and collectivism were associated with positive attitudes toward help-seeking, while individualism was associated with negative attitudes toward help-seeking. Interventions that normalize and destigmatize mental health challenges should be adapted to reflect the unique experiences of international students, and new interventions may seek to highlight the value of increased contact and collectivistic attitudes in facilitating mental health help-seeking.

## Introduction

International students in the United States (U.S.) are at increased risk of several mental health related challenges, including depression, anxiety, and associated academic stress [1–3]. In their systematic review on international student adjustment in the U.S. and other English-speaking countries, Alharbi and Smith [4] identified 38 studies highlighting unique, impactful experiences of stress that affect international student mental health. These included acculturation stress, language challenges, discrimination, loneliness, and academic stress. As a result, international students experienced increased mental health challenges compared to their U.S.-born peers [4].

Despite these unique and elevated experiences of mental health distress, international students are less likely than their U.S.-born peers to seek professional mental health support,

**Funding:** The authors received no specific funding for this work.

**Competing interests:** The authors have declared that no competing interests exist.

including services at university counseling centers [5–7]. Instead, international students are more likely to seek support from friends, family, and academic advisors when experiencing emotional distress [7]. Although these are valuable sources of support, these patterns of help-seeking can be problematic if they are viewed as a replacement or proxy for needed professional support [6]. Researchers have highlighted stigma toward mental health and help-seeking as key barriers for international students, which may be rooted in misinformation or negative cultural beliefs about mental health and its treatment [8]. However, with sound education and healthy exposure to mental health treatment, these attitudes may improve [5]. It is, therefore, critically important for researchers to conduct studies to understand the influence of stigma and other key barriers to professional counseling among international students.

During the COVID-19 pandemic, international students faced a multitude of stressors, some of which were unique to this population; these included social isolation, campus closures in an unfamiliar country, travel restrictions, and challenges navigating an unfamiliar medical system [9]. Even prior to the COVID-19 pandemic, acculturation and adjustment to the U.S. have been major factors influencing international student mental health [10]. Preliminary data show that the pandemic had a considerable negative impact on international student mental health, including depression and anxiety, which was exacerbated by academic and financial stress [11, 12]. Prior personal experience with mental illness and familiarity with mental illness have also been associated with less stigmatizing attitudes toward mental health help-seeking [13, 14].

Key theoretical approaches have emerged to facilitate understanding of international student mental health, including acculturation theory and social identity theory [5]. These frameworks acknowledge the interaction between unique stressors faced by international students and the normal developmental challenges that often occur in emerging adulthood [15]. Cultural background and country of origin may also influence mental health stigma and rates of help-seeking [8]. This includes one's level of familiarity with mental health treatment [10], attitudes toward people living with a mental illness [8, 16], and the degree of individualism and collectivism in one's culture of origin, all of which may inform whether, where, and how people seek emotional support [10, 17].

Few studies have used combined models to examine acculturation and social identity factors that may be associated with stigma and help-seeking among international students, particularly during the COVID-19 pandemic. In their valuable systematic review of studies focused on international student psychosocial adjustment, Brunsting and colleagues called for novel research with international students focused on "understanding primary relationships between student characteristics, their experiences, and their adjustment outcomes"[5, p. 31].

The objective of this exploratory study was to expand the literature on this understudied topic by exploring associations between demographics, time in the U.S., prior contact with mental illness, and individualism/collectivism with stigma and attitudes toward mental health help-seeking in a small but diverse sample of international students at U.S. colleges and universities during the COVID-19 pandemic. We hypothesized that time in the U.S., increased contact with mental illness, collectivist attitudes, and identifying as a woman would be associated with decreased stigma and more positive attitudes toward help-seeking in this population, while individualistic attitudes and religiosity would be associated with increased stigma and more negative attitudes toward help-seeking.

## Methods

We administered a one-time, cross-sectional online survey to 132 international students enrolled at U.S. colleges and universities between October 28, 2020 and March 11, 2022.

International students were defined as any student enrolled at the college or university who was not born in the U.S. and was recognized as an international student by their institution's International Student Services or equivalent office. To recruit participants, we first contacted International Student Services or the equivalent office at U.S. colleges and universities, requesting that they distribute the invitation to participate to international students at their institution, including both undergraduate and graduate students. We invited 126 colleges and universities to distribute the invitation and 14 (11.1%) ultimately agreed, representing 10 different geographically diverse U.S. states. The majority of institutions (10 institutions, 71.4%) were 4-year public universities, 2 (14.3%) were 4-year private universities, and 2 (14.3%) were 2-year community/technical colleges. Among the participating institutions, 8 distributed the survey invitation by email listserv, 4 by newsletter, and 2 by posting to institutional social media accounts.

Students were eligible to participate if they were an international student currently enrolled at a U.S. college or university, at least 18 years of age, and fluent in English. Interested students clicked on a hyperlink in the invitation and were directed to a secure survey website where they confirmed their eligibility and provided informed consent prior to proceeding to the online survey. The average student response rate at 4 institutions who provided data on the number of international students invited was 4.6% (range 4.4–8.9%), which falls within the normal range for surveys with Internet recruitment [18, 19]. A total of 212 individuals accessed the survey, but 80 provided blank or incomplete surveys, yielding our final sample of 132 participants.

On average, the survey took 23 minutes to complete. No identifying information was collected on the survey. At the conclusion of the survey, students could click a link to a second, unconnected site where they could sign up for a drawing to win one of ten $20 gift cards.

## Ethics statement

All participants confirmed their eligibility and provided informed consent by indicating "I agree" with the informed consent statement at the start of the online survey. All participants were 18 years of age or older. The study received ethical approval from the institutional review board at Duke University, protocol number 2021–0109.

## Measures

*Demographic Variables* included participant gender, age, race/ethnicity, country of origin, and religion. Participants also indicated the number of months they had been living in the U.S. A histogram of participant ages was skewed to the right with a clear bimodal distribution above and below the mean; therefore, age was dichotomized at the mean of 25 years and older. Given the inclusion of all students born outside of the U.S., the variable of months living in the U.S. showed an expected positive skew but did not have a clear bimodal distribution, so we completed a log transformation for this variable.

*Religious Importance* was measured by a single item, "How important is your religion to you?" Responses were provided on a 6-point Likert-type scale ranging from 0-Very Unimportant to 6-Very Important.

*Contact with Mental Illness* was measured by a modified Level-of-Contact report [20] with six yes/no questions of whether the personal had encountered people living with mental illness in their daily life, work, social networks, family, or experienced a mental illness themselves. Items are ordered ranging from least to most contact and scored accordingly, with "no" to all items representing a score of 0 and "yes" to personally experiencing a mental illness representing a maximum score of 6. The score is representative of the highest item endorsed, even if

lower items are not endorsed. For example, a person who has personally experienced a mental illness receives a score of 6, regardless of whether they have lived with or worked with a person experiencing mental illness.

*Individualism and Collectivism* were measured using the Individualism/Collectivism Survey [21], which consists of 16 items measured on a 9-point scale from 1-Strongly Disagree to 9-Strongly Agree. The developers of the measure identified individualism and collectivism as distinct variables that are not mutually exclusive and should be measured separately. Therefore, we summed the scores for two 8-item subscales, one for individualism and one for collectivism, each with a total score ranging from 1 to 72 and a higher score representing stronger individualistic or collectivistic beliefs.

*Mental Health Stigma* was measured by the Stigma Through Knowledge Test (STKT) [22], a 14-item faux knowledge test. On the study survey, this instrument is labeled, "Knowledge Test About Mental Illness", but the items are in fact used to measure stigmatizing attitudes about mental illness, such as beliefs that people with mental illness cannot recover from their symptoms, cannot live independently, or cannot benefit from treatment. For example, one true/false item is "People with severe mental illness cannot maintain private residences." We chose the STKT because the faux knowledge test format has been shown to successfully reduce tendencies toward choosing socially desirable responses, which is a common threat to the validity of traditional stigma measures [22]. Each STKT item has one stigmatizing response scored 1 and one non-stigmatizing response scored 0, yielding a total score of 0 to 14 with a higher score indicating more stigma.

*Attitudes Toward Mental Health Help-Seeking* was measured with the 10-item Attitudes Toward Seeking Professional Psychological Help–Short Form (ATSPPH-SF) [23]. Items describe the confidence, value, and need of seeking professional psychological help (e.g., "I would want to get psychological help if I was worried or upset for a long period of time.") and are scored on a 4-point Likert-type scale from 0-Disagree to 3-Agree, for a total range of 0 to 30 with higher scores indicating more positive attitudes toward help-seeking. A histogram of total scores on the ATSPPH-SF showed a clear bimodal distribution above and below the mean; therefore, scores were dichotomized at the mean. Scores of 21 and above showed positive help-seeking attitudes and those below 21 showed negative help-seeking attitudes.

## Statistical analysis

We used descriptive statistics to report the characteristics of the sample. We used linear regression to examine potential factors associated with mental health stigma as measured by the STKT, including participant gender, age, months in the U.S., religious importance, contact with mental illness, individualism, and collectivism. We then used binary logistic regression to examine potential factors associated with attitudes toward mental health help-seeking as measured by the dichotomized ATSPPH-SF, examining the same independent variables plus mental health stigma. Factors in the regression models were chosen based on prior studies showing these variables to be independently associated with stigma and help-seeking as described in the Introduction. P-values less than .05 were considered statistically significant. Missing data were imputed with the individual mean of completed items when at least 75% of items were completed, a robust method when both the number of participants missing data and the total number of missing items are small, as was the case in our data [24, 25]. Significant intercorrelations were common among the continuous independent variables but none exceeded preidentified thresholds for multicollinearity ($r > .80$ or VIF $> 5.0$) [26, 27].

## Results

### Participants

Participants included 132 international students at U.S. colleges and universities representing 51 unique nations of origin. The mean age of participants was 25 years old and participants had been in the U.S. for 35 months on average (median = 24), which reflects the inclusion of graduate and professional students and participants who were born outside of the U.S. but had lived in the U.S. prior to commencing their studies. See Table 1 for additional details about the characteristics of the study participants.

The overall mean stigma score in the sample was 6.26 (SD = 2.34) out of 14 possible on the STKT scale. The most frequently endorsed items on this faux knowledge test were "Neglectful parenting is somewhat responsible for the beginning of a serious mental illness" ($n = 100$, 75.8%) and "Adolescents with schizophrenia are frequently truant from school" ($n = 98$, 74.2%). In the linear regression model, only prior contact with mental illness was significantly associated with mental health stigma in both the univariable and multivariable models. Each 1-point increase in the level of prior contact with mental illness was associated with a .311 log count decrease in stigma in the univariable model ($p < .001$), and a .307 log count decrease in stigma in the multivariable model ($p = .003$). See Table 2 for additional details.

The overall mean score for Attitudes Toward Mental Health Help-Seeking in the sample was 20.51 (SD = 5.33) out of 30 possible on the ATSPPH-SF. On the dichotomized variable, 65

**Table 1. Characteristics of the study participants.**

| Category | Value |
|---|---|
| Total Participants | 132 |
| Gender, n (%) | |
| Women | 83 (63.8%) |
| Men | 47 (36.2%) |
| Age (mean, median) | 25, 25 |
| Race/Ethnicity, n (%) | |
| Asian | 64 (48.9%) |
| White | 30 (22.9%) |
| Hispanic or Latinx | 15 (11.5%) |
| Black or African | 11 (8.4%) |
| Middle Eastern or North African | 7 (5.3%) |
| More Than One | 4 (3.1%) |
| Months in U.S. (mean, median) | 35, 24 |
| Nation of Origin—Top 6 Listed, n (%) | |
| China | 28 (21.2%) |
| India | 11 (8.3%) |
| United Kingdom | 8 (6.1%) |
| Mexico | 6 (4.5%) |
| Italy | 5 (3.8%) |
| Religion | |
| None/Agnostic/Atheist | 57 (43.2%) |
| Christian | 47 (35.6%) |
| Muslim | 10 (7.6%) |
| Hindu | 8 (6.1%) |
| Buddhist | 6 (4.5%) |
| Other | 4 (3.1%) |

**Table 2. Factors associated with mental health stigma (N = 132).**

| | Stigma (mean) | Univariable B (95% CI) | Multivariable B (95% CI) |
|---|---|---|---|
| *Gender* | | | |
| Women | 6.2 | REF | REF |
| Men | 6.5 | .056 (-0.58, 1.12) | -.007 (-1.05, .980) |
| *Age* | | | |
| 25 years and above | 6.0 | REF | REF |
| < 25 years old | 6.6 | .144 (-.128, 1.48) | .129 (-.343, 1.55) |
| *Months in The U.S.* | | -.160 (-.567, .020) | -.102 (-.526, .175) |
| *Religious Importance* | | -.052 (-.292, .159) | .030 (-.234, .310) |
| *Contact with Mental Illness* | | -.311 (-.516, -.158)*** | -.307 (-.549. -.116)** |
| *Individualism* | | .051 (-.044, .074) | .000 (-.059., .059) |
| *Collectivism* | | .151 (-.014, .108) | .141 (-.021, .108) |

Note.

*$p \leq .05$.

**$p < .01$.

***$p < .001$. B, unstandardized beta. CI, confidence interval. REF, reference category.

participants (53.7%) met or exceeded the cutoff score of 21 for positive attitudes toward help-seeking. The most highly scored item on the ATSPPH-SF was "A person should work out his or her own problems; getting psychological counseling would be a last resort" (reverse scored, mean = 2.35/3.00) and the lowest scored item was "A person with an emotional problem is not likely to solve it alone; he or she is likely to solve it with professional help" (mean = 1.74/3.00).

In the binary logistic regression model, women were significantly more likely to endorse positive attitudes toward help-seeking as compared to male participants in both the univariable (OR = 3.10, $p = .005$) and multivariable models (OR = 4.54, $p = .009$). Prior contact with mental illness was also associated with more positive attitudes toward help-seeking in both the univariable (OR = 1.40, $p < .001$) and multivariable models (OR = 1.45, $p = .006$). Higher individualism was associated with negative attitudes toward help-seeking in both the univariable (OR = .934, $p = .015$) and multivariable models (OR = .888, $p = .002$). Collectivism was associated with more positive attitudes toward help-seeking in the multivariable model only (OR = 1.08, $p = .050$). See Table 3 for additional details.

## Discussion

Prior studies have posited that stigmatizing attitudes toward mental illness are a key driver of low help-seeking among international students [8, 28]. However, few studies have examined combined models including factors such as time in the U.S., religiosity, prior contact with mental illness, individualism, and collectivism as potential drivers of stigma and perceptions of mental health help-seeking among international students, particularly during the COVID-19 pandemic.

In this exploratory cross-sectional survey of international students in the U.S., the data supported our hypothesis that participants with more prior contact with mental illness—including knowing someone with a mental illness, having a family member with a mental illness, or experiencing a mental illness oneself–was associated with reduced mental health stigma. This finding reflects the broader stigma literature with other populations, including many studies with students [29, 30]; however, no known studies have examined associations of prior contact

**Table 3. Factors associated with attitudes toward mental health help-seeking (*N* = 132).**

| | Negative Attitudes *n* (%) | Positive Attitudes *n* (%) | Univariable Analysis, OR (95% CI) | Multivariable Analysis, OR (95% CI) |
|---|---|---|---|---|
| *Gender* | | | | |
| Men | 26 (63.4%) | 15 (36.6%) | REF | REF |
| Women | 28 (35.9%) | 50 (64.1%) | 3.10 (1.41, 6.79)** | 4.54 (1.46, 14.08)** |
| *Age* | | | | |
| <25 years old | 27 (48.2%) | 29 (51.8%) | REF | REF |
| 25 years and above | 29 (44.6%) | 36 (55.4%) | 1.16 (.564, 2.37) | .859 (.306, 2.41) |
| | Negative Attitudes, Mean (IQR) | Positive Attitudes, Mean (IQR) | Univariable Analysis, OR (95% CI) | Multivariable Analysis, OR (95% CI) |
| *Months in the U.S.* | 32.3 (4.00, 36.0) | 39.6 (12.0, 54.0) | 1.27 (.967, 1.67) | 1.08 (.736, 1.58) |
| *Religious Importance* | 3.39 (2.00, 5.00) | 3.47 (2.00, 5.00) | 1.02 (.840, 1.25) | 1.08 (.801, 1.47) |
| *Contact with Mental Illness* | 2.88 (0.00, 5.00) | 4.29 (3.00, 6.00) | 1.40 (1.16, 1.69)*** | 1.45 (1.11, 1.89)** |
| *Individualism* | 39.0 (34.5, 44.0) | 35.0 (29.0, 40.0) | .934 (.885, .987)* | .888 (.824, .956)** |
| *Collectivism* | 39.6 (34.0, 46.0) | 41.8 (38.0, 46.0) | 1.04 (.986, 1.10) | 1.08 (1.00, 1.16)* |
| *Mental Health Stigma* | 6.45 (5.00, 8.00) | 6.03 (4.00, 8.00) | .929 (.799, 1.08) | .986 (.777, 1.25) |

Note.

*$p \leq$ .05.

**$p <$ .01.

***$p <$ .001. OR, odds ratio. CI, confidence interval. REF, reference category. IQR, interquartile range.

with mental illness among international students specifically [31]. In broader college and university settings, there have been effective campaigns to educate students about mental illness. Encouraging students to speak out about their experiences with mental illness and mental health treatment serves the dual purpose of increasing contact with mental illness and reducing mental health stigma [13, 32, 33]. Our novel findings provide a strong rationale for conducting larger research studies to inform the development of programs targeted specifically to reduce mental health stigma among international students, focused on bringing personal experiences with mental illness out of the shadows and into the campus discourse.

No other factors in our linear regression model were significantly associated with mental health stigma, including variables such as gender, age, and time spent in the U.S., which have been associated with mental health stigma in prior studies [8, 18]. When considering reasons for these non-significant associations in the current analysis, one key difference in our study is the use of a faux knowledge test (the STKT) as opposed to a more overt measure of stigmatizing attitudes. It is possible that some groups of participants are more prone to providing socially desirable responses on traditional stigma measures. Another factor may be historical: that the international students in our study who are men, are younger, or who are newer to the U.S. may now be less prone to stigma than they were in the past due to global efforts in recent years to educate the public about mental health challenges, raise awareness, reduce stigma, and encourage mental health help-seeking [34, 35].

In examining factors associated with attitudes toward mental health help-seeking, our findings were more reflective of both our hypotheses and the broader literature. In our analysis, women had significantly more positive attitudes toward help-seeking. This is similar to studies with other college student populations, which found considerably higher help-seeking and treatment utilization among women [36, 37]. We also found that increased prior contact with mental illness was strongly associated with help-seeking. This may be due to increased sensitivity to the needs of people struggling with mental health challenges, or due to prior experiences

with treatment or as a treatment supporter [38]. Interestingly, stigma was not associated with mental health help-seeking in the current study, which differs from the broader literature [39] and may again be related to the use of a faux knowledge test to measure mental health stigma.

Individualism and collectivism have rarely been examined as correlates of mental health help-seeking, particularly among college students. In the current study, individualism was associated with more negative attitudes toward mental health help-seeking while collectivism was associated with more positive attitudes toward help-seeking. This supports prior studies' results by showing that perceptions of interpersonal independence and interdependence may extend beyond immediate social networks to influence perceptions of professional support and treatment [10, 40]. We provide new evidence that these attitudes are influential among international students. Future research should seek to assess whether such individualistic attitudes that prevent help-seeking are malleable in addition to exploration of interventions that highlight the value of collectivistic attitudes in facilitating help-seeking.

Time spent in the U.S. was not significantly associated with lower mental health stigma, nor with mental health help-seeking in the current study. These findings differ from the prior literature [8, 41]. The lack of significance in these relationships in the current study may indicate that stigma and help-seeking attitudes have become more stable in the early months and years after international students arrive in the U.S. Broader reductions in stigma and increases in acceptance of mental health treatment in other countries may be rendering these attitudes less prone to acculturative change [39]. Future studies may seek to assess longitudinal or intergenerational differences in acculturation related to mental health stigma and help-seeking as well as the potential influence of experiences as an international student on these acculturative processes.

The study findings should be interpreted in light of the following limitations. The institutional participation rate and student response rate were low, and the overall sample size is correspondingly low. Therefore, the results may not be representative of the broader international student population in the U.S. Estimates have not been weighted to the national population of international students due to lack of adequate data to do so. Future studies with larger samples may seek to explore sub-group differences within the international student populations based on, for example, country of origin and visa type. Additionally, it is possible that the exclusion of students not fluent in English influenced the study findings, although English fluency is required to attend nearly all higher education institutions in the U.S. Given these limitations, the findings of the study should be interpreted with caution.

## Conclusions

In this cross-sectional survey of international students in the U.S., we found that having more prior contact with mental illness was strongly associated with lower stigma and higher acceptance of mental health help-seeking. Identifying as a woman and collectivism were also associated with higher acceptance of help-seeking, while individualism was associated with lower acceptance of help-seeking. Successful interventions designed to normalize and destigmatize mental health challenges among university students should be adapted to reflect the unique experiences of international students, and new interventions may seek to highlight the value of collectivistic attitudes in facilitating help-seeking.

## Author Contributions

**Conceptualization:** Brandon A. Knettel, Pranav Ganapathy.

**Data curation:** Brandon A. Knettel, Pranav Ganapathy.

**Formal analysis:** Brandon A. Knettel, Pranav Ganapathy.

**Investigation:** Brandon A. Knettel, Pranav Ganapathy.

**Methodology:** Brandon A. Knettel, Pranav Ganapathy, Conner Rougier-Chapman.

**Project administration:** Brandon A. Knettel.

**Supervision:** Brandon A. Knettel.

**Visualization:** Brandon A. Knettel.

**Writing – original draft:** Brandon A. Knettel, Pranav Ganapathy, Conner Rougier-Chapman.

**Writing – review & editing:** Brandon A. Knettel, Pranav Ganapathy, Conner Rougier-Chapman.

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
