## [Decision Letter · Decision Letter 0]

15 Mar 2024

PMEN-D-24-00027

Examining factors associated with mental health stigma and attitudes toward help-seeking among international college students during the COVID-19 pandemic

PLOS Mental Health

Dear Dr. Knettel,

Thank you for submitting your manuscript to PLOS Mental Health. After careful consideration, we feel that it has merit but does not fully meet PLOS Mental Health’s publication criteria as it currently stands. Therefore, we invite you to submit a revised version of the manuscript that addresses the points raised during the review process.

We look forward to receiving your revised manuscript.

Kind regards,

Terry Fleming

Academic Editor

PLOS Mental Health

Journal Requirements:

1. Please send a completed 'Competing Interests' statement, including any COIs declared by your co-authors. If you have no competing interests to declare, please state "The authors have declared that no competing interests exist".   2. Please amend your detailed Financial Disclosure statement. This is published with the article. It must therefore be completed in full sentences and contain the exact wording you wish to be published. If you did not receive any funding for this study, please simply state: “The authors received no specific funding for this work.”"  3. In the online submission form, you indicated that "
Permission to share the full data set was not obtained from the participating institutions at the time of data collection. Requests for de-identified data may be made available upon request from the authors, or from the Duke University Center for Nursing Research, with an appropriate data transfer agreement.".   All PLOS journals now require all data underlying the findings described in their manuscript to be freely available to other researchers, either 1. In a public repository, 2. Within the manuscript itself, or 3. Uploaded as supplementary information. This policy applies to all data except where public deposition would breach compliance with the protocol approved by your research ethics board. If your data cannot be made publicly available for ethical or legal reasons (e.g., public availability would compromise patient privacy), please explain your reasons by return email and your exemption request will be escalated to the editor for approval. Your exemption request will be handled independently and will not hold up the peer review process, but will need to be resolved should your manuscript be accepted for publication. One of the Editorial team will then be in touch if there are any issues.  4. Please cite the tables within the main text/paragraph.  

Additional Editor Comments (if provided):

Thank you for your submission, please address the reviewers comments. Note all reviewers had comments about your definition of international students and the lack of consideration of limitations of the study. Please address each comment and resubmit for re-review

Best wishes Terry Fleming

Reviewers' comments:

Reviewer's Responses to Questions

**Comments to the Author**

1. Does this manuscript meet PLOS Mental Health’s publication criteria? Is the manuscript technically sound, and do the data support the conclusions? The manuscript must describe methodologically and ethically rigorous research with conclusions that are appropriately drawn based on the data presented.

Reviewer #1: Yes

Reviewer #2: Yes

Reviewer #3: Partly

2. Has the statistical analysis been performed appropriately and rigorously?

Reviewer #1: Yes

Reviewer #2: Yes

Reviewer #3: Yes

3. Have the authors made all data underlying the findings in their manuscript fully available (please refer to the Data Availability Statement at the start of the manuscript PDF file)?

Reviewer #1: Yes

Reviewer #2: Yes

Reviewer #3: No

4. Is the manuscript presented in an intelligible fashion and written in standard English?

Reviewer #1: Yes

Reviewer #2: Yes

Reviewer #3: Yes

5. Review Comments to the Author

Reviewer #1: Recommendation: Minor revision

This is a carefully conducted original study and a well-written paper which adds novel findings to the literature on mental health stigma and help-seeking. The findings have the potential to inform practice in relation to supporting international students in the US and internationally. The findings are discussed appropriately in the context of the existing literature, and the authors are clear about what this study adds.

My only major concern is that the limitations of the study – which are significant - are not highlighted in the discussion. The institutional participation rate and student response rate were low, and the overall sample size is correspondingly low, so the results may not be representative. Estimates have not been weighted to the national population of international students – do data exist to allow this? Could the exclusion of students not fluent in English introduced bias?

Given the study's limitations, the findings should be interpreted with caution. For example on P 14 “…by showing that perceptions of interpersonal independence and interdependence MAY extend…”

I would also like to query the definition of an ‘international student’ i.e. anyone not born in the USA. In a country like the USA with high immigration, surely a high proportion of university students were born elsewhere but would not be considered ‘international students’? The average time spent in the US suggests your survey did NOT pick up people who had spent most of their lives in the US, and I presume this is because the survey was distributed via International Student Services. Perhaps a clearer definition would be “students under the care of International Student Services or equivalent AND not born in the US” or similar?

Reviewer #2: Thank you for the opportunity to review this well written manuscript.

I have a few small points that could be further clarified, my apologies for engaging beyond the material provided:

Introduction:

Please make clear if Alharbi and Smith's work covers US research.

Please also define all acronyms before using them (US).

Can you adjust the sentence "Although these are valuable sources of support, these patterns of help-seeking can be problematic when they are relied upon instead of, rather than in addition to, professional support." At the moment, it appears to suggest that professional support should be used instead of family/ friend support, which may not be appropriate/ necessary in the first instance.

"It is therefore critically important for studies to understand the influence of stigma and other key..." Add a comma after therefore, and also review the sentence as studies do not understand, people do.

Method:

In reading that international students were defined as those not born in the US, I wonder what proportion would consider themselves to be international students. Some could have grown up in the US or have significant familial connections here. While it appears the average student in this study has been in the country for only a few years, in an increasingly globalised world it is still likely that they may have connections in country.

A statement around what level of statistical significance was used would also be beneficial.

Similarly, explaining what % of surveys were excluded for being incomplete would strengthen this study, as would having a range for the response rate between the universities.

The method also lacks explanation for how the response rate was calculated - did the authors know the number of international students in each university?

When considering individualism vs. collectivism, was there any attempt made to identify cultures that are considered in the research to be more collective (rather than solely looking at the individual)?

Was data collected on the form of visa students had when studying e.g., refugee populations may face additional barriers than other international students.

Results:

Given that both undergraduate and postgraduate students were surveyed, I find myself wondering if there were differences based on this, or indeed the course of study undertaken.

In Table 1, please change "Christian/ Catholic" to Christian - it doesn't make sense to identify this denomination but not others. In addition, please separate atheist from none, as atheists believe that god does not exist (which in itself could be considered holding a stance more closely aligned with religious groups).

In the paragraph before table 2, the authors write "Each 1-point increase in the level of prior contact with mental illness was associated with a .311 log count decrease in stigma in the univariable model (p<.001), and a .307 log count decrease in stigma in the multivariable model (p=.003)." This sentence should then lead on to a statement about the implication/ meaning of this.

Discussion:

I would have liked a greater focus on potential study limitations. For example, acknowledgement of the low response rate. Were the universities who participated different in any way from those that did not? Did they have programmes to support international students?

Perhaps it is my discipline, but I am inclined to ask for additional tangible recommendations on how to apply the research findings. At present, this is perhaps the main weakness of the study.

Reviewer #3: This is an important area of study, which has not been adequately addressed in literature. However, the manuscript can be improved.

Introduction

The authors mention that “During the COVID-19 pandemic, international students faced a multitude of stressors, some of which were unique to this population; these included social isolation, campus closures in an unfamiliar country, travel restrictions, and challenges navigating an unfamiliar medical system. Even prior to the COVID-19 pandemic, acculturation and adjustment to the United States have been major factors influencing international student mental health”. Can the authors go on to describe how the mental health of international students have changed during the pandemic?

Methods

The authors define an international student as “any student enrolled at the college or university who was not born in the U.S”. This is not in keeping with the widely accepted definition of an international student. The authors definition of international students can include students who were not born in USA, but have migrated to USA when they were very young, and are current citizens or permanent residents of USA. Can those who are not born in USA, but are current citizens/permanent residents of USA be labelled as international students? Were they excluded from the study? If so, this should be mentioned. If not, the authors should give justification why they were included as international students. The experience of those who migrated to USA when they were very young and entered USA collage/university after completing primary and secondary education in USA is very different from those who arrive in USA for tertiary education for the first time. Did the authors exclude those who have had primary/secondary education in USA? Can the authors please explain this in greater detail as it seems to be confusing.

Discussion

When giving reasons for not having any associations between demographic factors and stigma, the authors state “another factor may be historical: that the international students in our study who are men, are younger, or who are newer to the U.S. may now be less prone to stigma than they were in the past”. It is not clear what is meant by this.

The authors should discuss why they used the definition “not born in the USA” to define international students, as it seems to be different from the definition of an international student in most universities in the USA.

6. PLOS authors have the option to publish the peer review history of their article (what does this mean?). If published, this will include your full peer review and any attached files.

**Do you want your identity to be public for this peer review?** For information about this choice, including consent withdrawal, please see our Privacy Policy.

Reviewer #1: No

Reviewer #2: **Yes: **Tara N Officer

Reviewer #3: No

---

## [Editor Report · Decision Letter 1]

16 Apr 2024

PMEN-D-24-00027R1

Examining factors associated with mental health stigma and attitudes toward help-seeking among international college students during the COVID-19 pandemic

PLOS Mental Health

Dear Dr. Knettel,

Thank you for submitting your manuscript to PLOS Mental Health. After careful consideration, we feel that it has merit but does not fully meet PLOS Mental Health’s publication criteria as it currently stands. Therefore, we invite you to submit a revised version of the manuscript that addresses the points raised during the review process.

We look forward to receiving your revised manuscript.

Kind regards,

Terry Fleming

Academic Editor

PLOS Mental Health

Journal Requirements:

1. Please amend your online Financial Disclosure statement. If you did not receive any funding for this study, please simply state: “The authors received no specific funding for this work.”

2. Please update your online Competing Interests statement. If you have no competing interests to declare, please state: “The authors have declared that no competing interests exist.”

Additional Editor Comments (if provided):

Thank you for your resubmission.

Most of the reviewer comments have been addressed. Please make the following minor adjustments to ensure that all of these points have been covered well.

The limitations of the study have been outlined in a section in the discussion. However, as you report, this is a relatively small study with relatively low response rates. These limitations mean the findings should be considered valuable and of interest but not definitive. The wording in other parts of the paper such the abstract, the statement of main findings (in the 2nd paragraph of the discussion) and your conclusion should reflect this. Please ensure you do not overstate the strength of the findings and implications in these places.

The decision you prefer re "Christian/Catholic" is not standard. Please reconsider this or add a short footnote to explain why you have done this at the bottom of the table.

Best wishes Terry Fleming
---

## [Editor Report · Decision Letter 2]

15 May 2024

Examining factors associated with mental health stigma and attitudes toward help-seeking among international college students during the COVID-19 pandemic

PMEN-D-24-00027R2

Dear Dr. Knettel,

We are pleased to inform you that your manuscript 'Examining factors associated with mental health stigma and attitudes toward help-seeking among international college students during the COVID-19 pandemic' has been provisionally accepted for publication in PLOS Mental Health.

Best regards,

Terry Fleming

Academic Editor

PLOS Mental Health